# Gas Phase Photocatalytic $CO_2$ Reduction, "A Brief Overview for Benchmarking"

**Shahzad Ali** [1],[†]**, Monica Claire Flores** [1],[†]**, Abdul Razzaq** [2],[†]**, Saurav Sorcar** [1]**,
Chaitanya B. Hiragond** [1]**, Hye Rim Kim** [1]**, Young Ho Park** [1]**, Yunju Hwang** [1]**, Hong Soo Kim** [1]**,
Hwapyong Kim** [1]**, Eun Hee Gong** [1]**, Junho Lee** [1]**, Dongyun Kim** [1]** and Su-Il In** [1],[*]

[1]  Department of Energy Science & Engineering, DGIST, 333 Techno Jungang-daero, Hyeonpung-eup, Dalseong-gun, Daegu 42988, Korea
[2]  Department of Chemical Engineering, COMSATS University Islamabad, Lahore Campus, 1.5 KM Defence Road, Off Raiwind Road, Lahore 54000, Pakistan
[*]  Correspondence: insuil@dgist.ac.kr; Tel.: +82-053-785-6417
[†]  These authors contributed equally to this work.

**Abstract:** Photocatalytic $CO_2$ reduction is emerging as an affordable route for abating its ever increasing concentration. For commercial scale applications, many constraints are still required to be addressed. A variety of research areas are explored, such as development of photocatalysts and photoreactors, reaction parameters and conditions, to resolve these bottlenecks. In general, the photocatalyst performance is mostly adjudged in terms of its ability to only produce hydrocarbon products, and other vital parameters such as light source, reaction parameters, and type of photoreactors used are not normally given appropriate attention. This makes a comprehensive comparison of photocatalytic performance quite unrealistic. Hence, probing the photocatalytic performance in terms of apparent quantum yield (AQY) with the consideration of certain process and experimental parameters is a more reasonable and prudent approach. The present brief review portrays the importance and impact of aforementioned parameters in the field of gas phase photocatalytic $CO_2$ reduction.

**Keywords:** apparent quantum yield; organic contaminations; photocatalyst; solar; $CO_2$ reduction; photoreactors

## 1. Introduction

Photocatalytic products, as a consequence of $CO_2$ photoreduction, are industrially desirable with the additional benefit of normalizing anthropogenic $CO_2$ [1]. It is inevitable to develop and design efficient photoreactors and optimize the photoreaction conditions in order to scale up the photocatalytic process, all in congruence with synthesis of robust photocatalysts [2]. Although there have been many studies pertaining to the synthesis of stable and efficient photocatalysts, only few studies are dedicated upon reaction engineering so as to ascertain the optimum reaction conditions and photo reactor design [3]. Both of these factors have significant influence on photocatalytic yield. For instance, product yield will be different for photoreactors with batch and continuous flow operation under different conditions of reaction parameters, feed type and concentration ratio, photocatalyst loadings, and sources of irradiation [4,5]. In particular, irradiation sources and its mode of irradiation over the photocatalyst have vital importance. As most of the frequently used photocatalysts are Ultraviolet (UV) light active, irradiation from UV source as compared to solar light will significantly enhance the product yield [6–8]. Further, solar concentrator technology with utilization of Fresnel lens for photocatalyst irradiation leads to enhanced light intensity and photon flux, which in turn renders a significant enhancement in yield of hydrocarbon products [8,9]. Although, most of the photocatalytic

reactions are carried out at room temperature, concentrating solar light will increase the temperature of the system, which will induce the thermal effect that alters the yield [6,7].

Another decisive factor is the exposed area of the photocatalyst to interact with the light irradiations. Even for the different types of photocatalysts, when tested under similar conditions and with yields reported per gram of photocatalyst, a comparison can be misleading because the exposure area, per gram of the photocatalyst, to light will be different for powder catalysts and thin films [10,11]. Moreover, irrespective of photoreaction under standard conditions, yields have been reported in different customary units i.e., ppm $cm^{-2}$ $h^{-1}$, $\mu mol$ $g^{-1}$ $h^{-1}$ and $\mu mol$ $cm^{-2}$ $h^{-1}$ which further complicates the comparison [12–15]. Thus, photocatalysts that are tested under such different conditions and reporting consequent yield in different units lead to ambiguity for fair performance comparison. It is inappropriate to compare the activities of the photocatalysts on the basis of their intrinsic performance without considering effects of optimum conditions and reactor geometries because of these implications.

The aforementioned bottlenecks strongly advocate the necessity for standardized protocols in order to compare the results for photocatalytic $CO_2$ reduction on equal grounds. Therefore, reporting yields on the basis of photonic efficiency, which incorporates the radiation source and exposure area of the photocatalyst, is an appealing approach [16]. This review specifically focuses on the parameters that influence the actual yield of the photocatalytic $CO_2$ reduction reactions and outlines the standard testing practices for comparison and evaluation of performance. Moreover, we have calculated the apparent quantum yield (AQY) of different research works, on the basis of data available, for comparing the efficiencies of the photocatalysts.

## 2. Role of Organic Contaminations

Photocatalysts synthesis predominately involves organic materials as reaction reagents. The residues of such organic materials are not easily removed, even with calcination at higher temperatures. Photocatalytic $CO_2$ reduction products mainly hydrocarbons, using such catalysts, originate concomitantly from these organic residues and $CO_2$ as well [1–3]. Thus, yields from such photocatalysts are overestimated when compared to the actual yield originating from photocatalytic $CO_2$ reduction. However, isotopic labelling by $^{13}CO_2$ is carried out to rule out the possible involvement of these organics in overall photocatalytic yield. To understand the effects of organic contamination, Yang et al. performed Fourier Transform Infrared Spectroscopy (FTIR) investigations for photocatalytic $CO_2$ reduction over $Cu(I)/TiO_2$ photocatalyst synthesized by two different ways: one with polyethylene glycol (PEG) and another without PEG. The FTIR results showed that $Cu(I)/TiO_2$ with PEG produces more CO (photocatalytic reduction product) as compared to $Cu(I)/TiO_2$ without PEG as shown in Figure 1. This CO originates from organic contaminants even without the introduction of $CO_2$. The possible mechanism of formation of CO is represented by Equations (1) and (2),

$$CO_2 + C \rightarrow 2CO \tag{1}$$

$$H_2O + C \rightarrow CO + H_2 \tag{2}$$

Calcination at high temperature and illumination under dry He/Ar is incapable of completely wiping out these contaminants [4,5]. However, UV treatment in the presence of $H_2O$ vapors is propitious in removing these contaminants [4]. Furthermore, they extended this study to elucidate the effects of prolonged and repeated pretreatment, for four cycles (7 h each) under He/$H_2O$, for abolishing organic contaminants. In another study involving the synthesis of Ti-SBA-15 while using P123 (Pluronic acid) and TEOS (Tetraethylorthosilicate) by calcining at 550 °C for 6 h, they revealed the generation of significant amounts of hydrocarbon products ($CH_4$, $C_2H_6$, $C_2H_4$) under He/$H_2O$ environment. However, the yield of such photocatalytic products was significantly decreased after the first cycle but not diminished completely as shown in Figure 2a–d [5]. Moreover, the control experiments for photocatalytic reduction of CO with $H_2O$ exhibited transition in enhanced selectivity from $C_1$ to $C_2$. Thus, CO produced as

a consequence of organic contamination, as shown in Equations (1) and (2) has profound effects over yield and selectivity of photocatalytic $CO_2$ reduction [5].

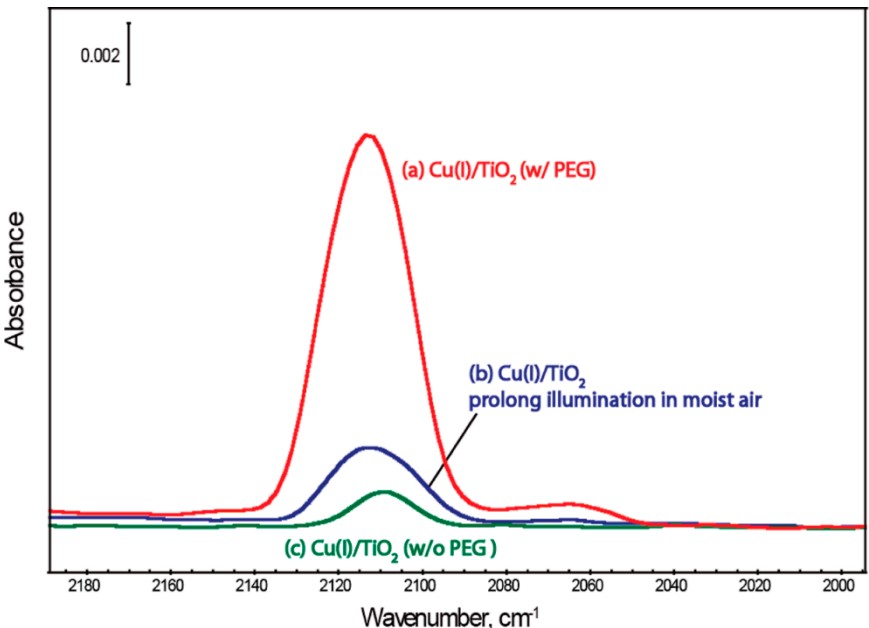

**Figure 1.** Fourier Transform Infrared Spectroscopy FTIR spectra of Cu(I)/TiO$_2$ preloaded with $^{13}CO_2$ after 80-min illumination. (**a**) fresh Cu(I)/TiO$_2$ (synthesized with PEG), (**b**) Cu(I)/TiO$_2$ cleaned by illumination in humid air for 14 h, and (**c**) reference Cu(I)/TiO$_2$ (synthesized without polyethylene glycol (PEG)), reproduced with permission from reference [4]. Copyright American Chemical Society, 2010.

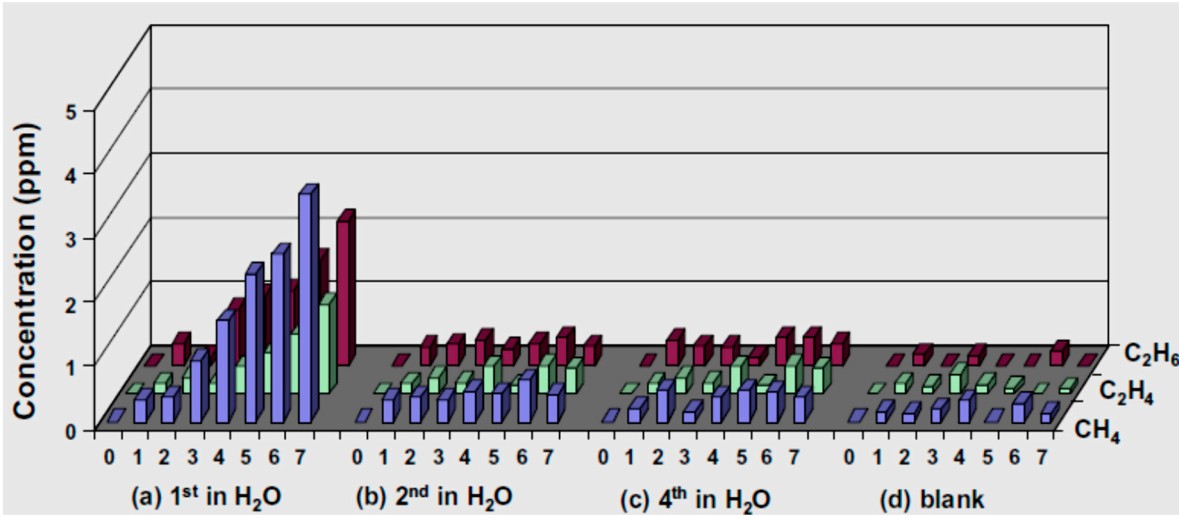

**Figure 2.** (**a–c**) production of CH$_4$, C$_2$H$_6$, and C$_2$H$_4$ over Ti-SBA-15 for four cycles, and (**d**) blank test (without catalyst). After every cycle reactor was evacuated with He/H$_2$O, reproduced with permission from reference [5]. Copyright Elsevier, 2011.

In a similar study by Busser et al., they reported the adsorption of $CO_2$ over the surface of the photocatalyst during photodeposition of Cu/Cr upon Ga$_2$O$_3$ in the presence of CH$_3$OH [1]. Their study confirmed that the $CO_2$ originates from photooxidation of methanol as shown in Equation (3).

$$Cu^{2+} + CH_3OH + H_2O \rightarrow Cu^0 + CO_2 + 6H^+ \tag{3}$$

Thus, the assessment of possible contribution of these organic contaminations is indispensable for gauging the actual yield.

## 3. Flow versus Batch Reactors

Photocatalysts that were tested under different reaction parameters and/or in different reactor geometries can have variability in reaction rate, yield and selectivity [6,7]. Literature suggests variety of reaction conditions and set-ups for photocatalytic $CO_2$ reduction. Of these conditions, two type of reaction modes are extensively applied in photocatalytic $CO_2$ reduction: (i) flow reaction system and (ii) batch reaction system [5,9–12]. However, the largely reported use of batch reactors obtains smaller yields of the photocatalytic products. Thus, the activity of the photocatalysts, tested under these different conditions, is imprecise to compare [2,10,13–17].

For batch reactors, it is difficult to understand and control the reaction mechanism and also product composition as well. This is because the products generated in the photocatalytic reaction may get re-adsorb over the surface of the photocatalysts or they can participate in side reactions to yield different products [9,18]. The key limitations of the batch reactor system include an accumulation of the products inside the reactor for a certain defined time which can lead to changes in the concentration of reactants by photocatalytic reactions itself, re-adsorption of the intermediate species or products, and the initiation of side reactions such as hydrogenation or re-oxidation to $CO_2$. For instance, $O_2$ that is produced during photocatalytic reduction when adsorbed on the surface of the photocatalyst might compete with $CO_2$ for electron intake. Thus, batch reactors for $CO_2$ photoreduction under various experimental conditions make it very difficult to compare the photocatalytic performance and they are not a suitable option for extended time and for industrial scale applications [12]. To resolve such issues, Pipelzadeh et al. designed a pressure swing reactor for facilitated product (CO mainly) desorption. In their configuration, the products were continuously recycled, thus periodic injection/evacuation of gases generated turbulence, which enhanced the CO production yield to 30–80%, depending upon flow rates as shown in Figure 3. Thus the continuous desorption of products is imperative which may surmount mass transfer limitations and the deactivation problems of the photocatalysts [19].

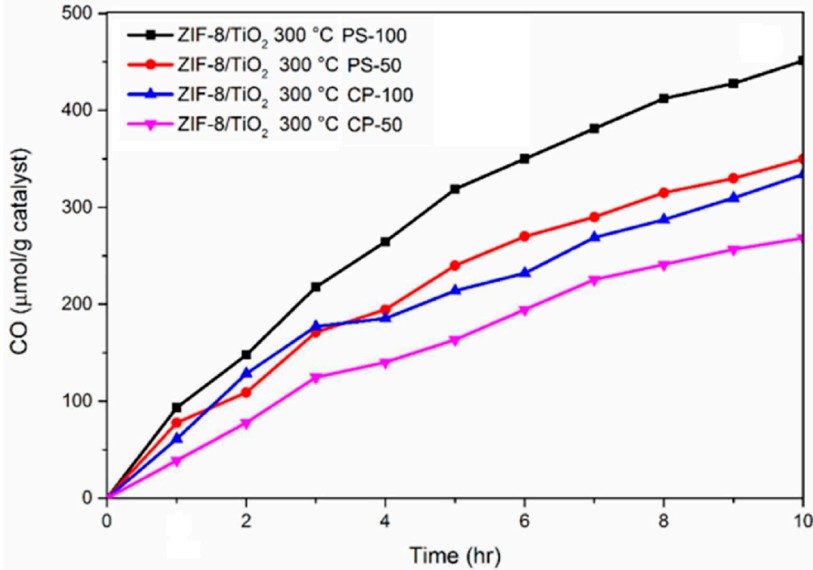

**Figure 3.** CO production for ZIF-8/$TiO_2$ under constant pressure (CP, 5 bar) and pressure swing (PS, 5–3 bar) at flow rates of 50 and 100 mL $min^{-1}$, reproduced with permission from reference [19]. Copyright Elsevier, 2017.

Contrary to batch reactors the re-adsorption of products and other aforementioned issues can be curtailed in continuous flow reactor system [9]. Despite that, the yield reported is still inadequate

since these types of systems only allow for short residence time of reactants i.e., restricting reactants to make proper contact with photocatalysts. However, better performance can be obtained by optimizing the reaction conditions and using robust photocatalysts [20,21]. In another study, a twin reactor, as shown in Figure 4, was designed to avoid the possible re-oxidation of the photocatalytic products. The photocatalytic $CO_2$ reduction results revealed that the production of hydrogen and oxygen in separate compartments and then use of as produced hydrogen in $CO_2$ reduction increased the yield of the products [22].

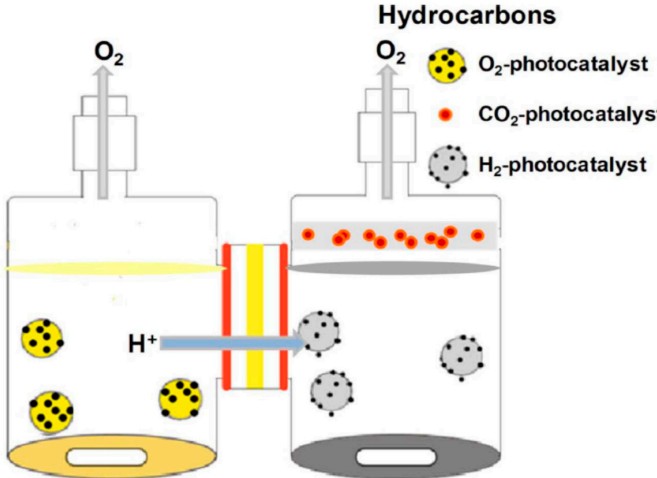

**Figure 4.** Illustration of twin reactor setup, reproduced with permission from reference [22]. Copyright Elsevier, 2018.

As evident from Table 1, studies by In et al. clearly vindicate the efficacy of transition from batch to continuous flow reactor along with the use of robust photocatalysts. Their well-designed flow reaction system, as shown in Figure 5 consists of a mass flow controller (50 standard $cm^3$, 20 °C, 1 atm) that regulates the flow of gases, a water bubbler (diameter = 3 cm, length = 15 cm) to maintain the desired humidity, a vacuum pump (pumping speed = 100 L $min^{-1}$) to obtain high purity conditions by degassing the ultra-sealed photoreactor under a vacuum of $3.5 \times 10^{-3}$ torr, and a gas chromatography unit for an automatic product intake and analysis (Shimadzu GC-2014, Restek Rt-Q Bond column, internal diameter = 0.53 mm, length = 30 m). Moreover, they used a ceramic porous disc (pore size = 1~1.6 µm) to support the catalyst and a quartz glass (diameter = 5 cm, thickness = 2 mm) to ensure efficient light transfer and the sealing of the reactor. Their research group also optimized the photoreactor dimensions to further improve the reaction process, (as shown in Figure 6a–c).

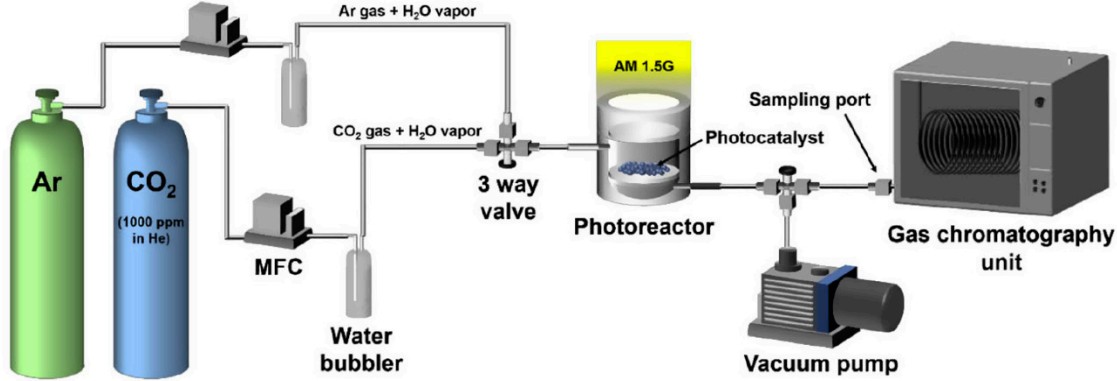

**Figure 5.** Setup for continuous gas phase photocatalytic $CO_2$ reduction, reproduced with permission from reference [11]. Copyright Elsevier, 2017.

**Table 1.** Summary of operating parameters and yields reported by In et al. for photocatalytic $CO_2$ reduction.

| Photocatalyst | Synthesis Method | Pre-Treatment of Reactor | Light Source | Reducing Agent | Reaction Parameters | Reactor | Main Product | AQY (%) |
|---|---|---|---|---|---|---|---|---|
| Degussa P25 standard titania [23] | store-bought | purged with high purity $CO_2$ gas, at least five times | UVP, UVGL-58 lamp with $\lambda = 365$ nm; 1200 $\mu$W cm$^{-2}$ | $H_2O$ | 50 mg catalyst on a 30 mm diam. glass disk; 15.4 cm$^3$ reactor; $CO_2$ flowrate @ 10 cm$^3$ min$^{-1}$; 500 $\mu$L sample gas extracted; ambient temperature and pressure; 1 h irradiation | Batch reactor | $CH_4$ @ 0.021 $\mu$mol g$^{-1}$ h$^{-1}$ | 0.0021 |
| CZTS−$TiO_2$ hybrid mesoporous [24] | hot injection and annealing | purged with $CO_2$ gas (1000 ppm in He) and vacuum | 100 W Xe solar simulator with an AM 1.5 filter; 100 mW cm$^{-2}$ | $H_2O$ | 50 mg catalyst on a 30 mm diam. glass disk; 15.4 cm$^3$ reactor; $CO_2$ flowrate @ 10 cm$^3$ min$^{-1}$; 500 $\mu$L sample gas extracted; ambient temperature and pressure; 1 h irradiation | Batch reactor | $CH_4$ @ 118.75 ppm g$^{-1}$ h$^{-1}$ | 0.0057 |
| CZTS-ZnO nanoparticles [14] | hydrothermal treatment | three times purged with $CO_2$ gas (1000 ppm in He) and vacuum | 100 W Xe solar simulator with an AM 1.5 filter; 100 mW cm$^{-2}$ | $H_2O$ | 50 mg catalyst on a 30 mm diam. glass disk; 15.4 cm$^3$ reactor; $CO_2$ flowrate @ 10 cm$^3$ min$^{-1}$; 500 $\mu$L sample gas extracted every 1 h; ambient temperature and pressure; 1 h irradiation | Batch reactor | $CH_4$ @ 0.0954 $\mu$mol g$^{-1}$ h$^{-1}$ | 0.0128 |
| $Cu_xO$−$TiO_2$ mesoporous p-type/n-type heterojunction material [25] | thermal decomposition then calcination | purged with $CO_2$ gas (1000 ppm in He) and vacuum | 100 W Xe solar simulator with an AM 1.5 filter; 100 mW cm$^{-2}$ | $H_2O$ | 50 mg catalyst on a 30 mm diam. glass disk; 15.4 cm$^3$ reactor; $CO_2$ flowrate @ 10 cm$^3$ min$^{-1}$; 500 $\mu$L sample gas extracted; ambient temperature and pressure; 1 h irradiation | Batch reactor | $CH_4$ @ 221.63 ppm g$^{-1}$ h$^{-1}$ | 0.0177 |
| Pt-x-RT nanoparticles [15] | magnesio-thermic reduction | five times purged with $CO_2$ gas (1000 ppm in He) and vacuum | 100 W Xe solar simulator with an AM 1.5 filter; 100 mW cm$^{-2}$ | $H_2O$ | 70 mg catalyst on a 30 mm diam. glass disk; 15.4 cm$^3$ reactor; $CO_2$ flowrate @ 10 cm$^3$ min$^{-1}$; 500 $\mu$L sample gas extracted; ambient temperature and pressure; 1 h irradiation | Batch reactor | $CH_4$ @ 1.13 $\mu$mol g$^{-1}$ h$^{-1}$ | 0.1234 |

**Table 1.** *Cont.*

| Photocatalyst | Synthesis Method | Pre-Treatment of Reactor | Light Source | Reducing Agent | Reaction Parameters | Reactor | Main Product | AQY (%) |
|---|---|---|---|---|---|---|---|---|
| C,N-TNT06 nanotubes [26] | alkaline hydrothermal technique then calcination | purged with $CO_2$ gas (1000 ppm in He) and vacuum | 100 W Xe solar simulator with an AM 1.5 filter; 100 mW cm$^{-2}$ | $H_2O$ | 100 mg catalyst on a 30 mm diam. glass disk; 15.4 cm$^3$ reactor; $CO_2$ flowrate @ 10 cm$^3$ min$^{-1}$; 500 μL sample gas extracted; ambient temperature and pressure; 1 h irradiation | Batch reactor | $CH_4$ @ 9.75 μmol g$^{-1}$ h$^{-1}$ | 1.0532 |
| Pt-XG/RBT nanoparticles [8] | facile vacuum treatment and photodeposition | 1 h purging with moist $CO_2$ gas @ 40 mL min$^{-1}$ | 100 W Xe solar simulator with an AM 1.5 filter; 100 mW cm$^{-2}$ | $H_2O$ | 40 mg catalyst on a 4.9 cm$^2$ porous disk; 26.57 cm$^3$ reactor; $CO_2$ flowrate @ 1 mL min$^{-1}$; sample gas analyzed every 30 min; ambient temperature and pressure; 7 h irradiation | Continuous flow reactor | $CH_4$ @ 37.0 μmol g$^{-1}$ h$^{-1}$ (AQY$_{CH4}$ = 5.248) $C_2H_6$ @ 11.0 μmol g$^{-1}$ h$^{-1}$ (AQY$_{C2H6}$ = 2.73) | 7.978 |
| Pt-BT-X nanoparticles [11] | facile low-temperature synthesis, annealing and photodeposition | 1 h purging with moist $CO_2$ gas @ 40 mL min$^{-1}$ | 100 W Xe solar simulator with an AM 1.5 filter; 100 mW cm$^{-2}$ | $H_2O$ | 40 mg catalyst on a 4.9 cm$^2$ porous disk; 26.57 cm$^3$ reactor; $CO_2$ flowrate @ 1 mL min$^{-1}$; sample gas analyzed every 30 min; ambient temperature and pressure; 6 h irradiation | Continuous flow reactor | $CH_4$ @ 80.35 μmol g$^{-1}$ h$^{-1}$ | 12.357 |
| Cu$_{x\%}$–Pt$_{y\%}$–BT nanoparticles [27] | facile low-temperature synthesis, annealing and photodeposition | 1 h purging with moist $CO_2$ gas @ 40 cc min$^{-1}$ | 100 W Xe solar simulator with an AM 1.5 filter; 100 mW cm$^{-2}$ | $H_2O$ | 40 mg catalyst on a 2.5 cm diam. porous disk; 26.57 cm$^3$ reactor; $CO_2$ flowrate @ 1 mL min$^{-1}$; sample gas analyzed every 30 min; ambient temperature and pressure; 6 h irradiation | Continuous flow reactor | $CH_4$ @ 3.0 mmol g$^{-1}$ (AQY$_{CH4}$ = 79.14) $C_2H_6$ @ 0.15 mmol g$^{-1}$ (AQY$_{C2H6}$ = 6.92) | 86 |

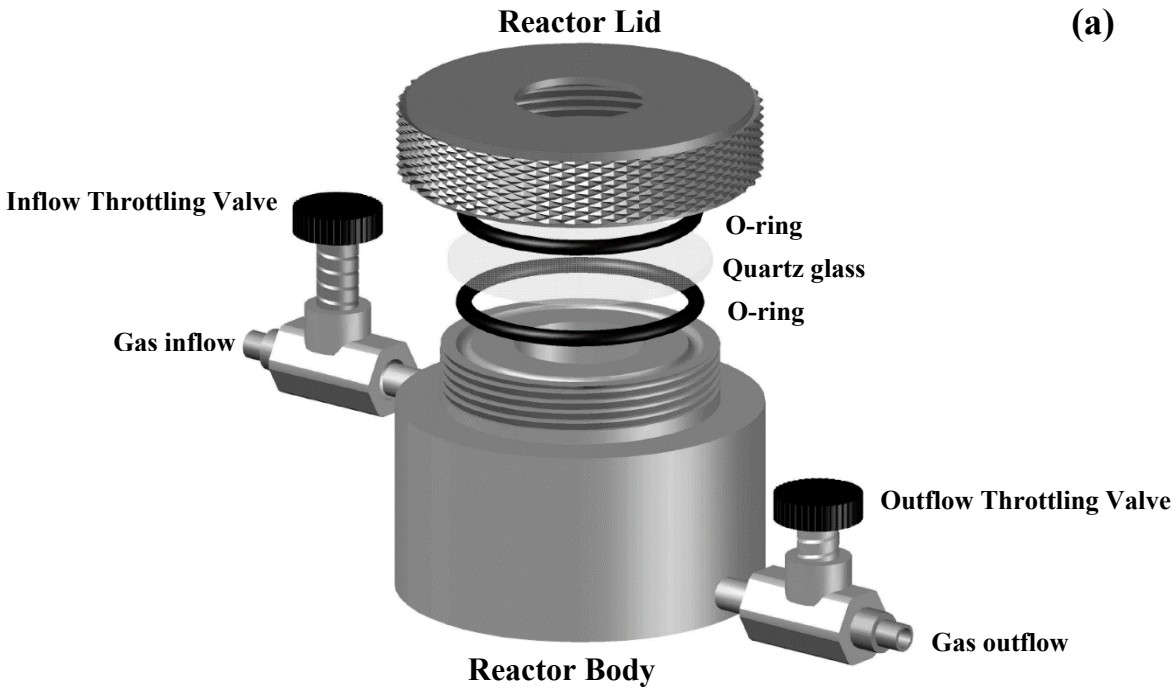

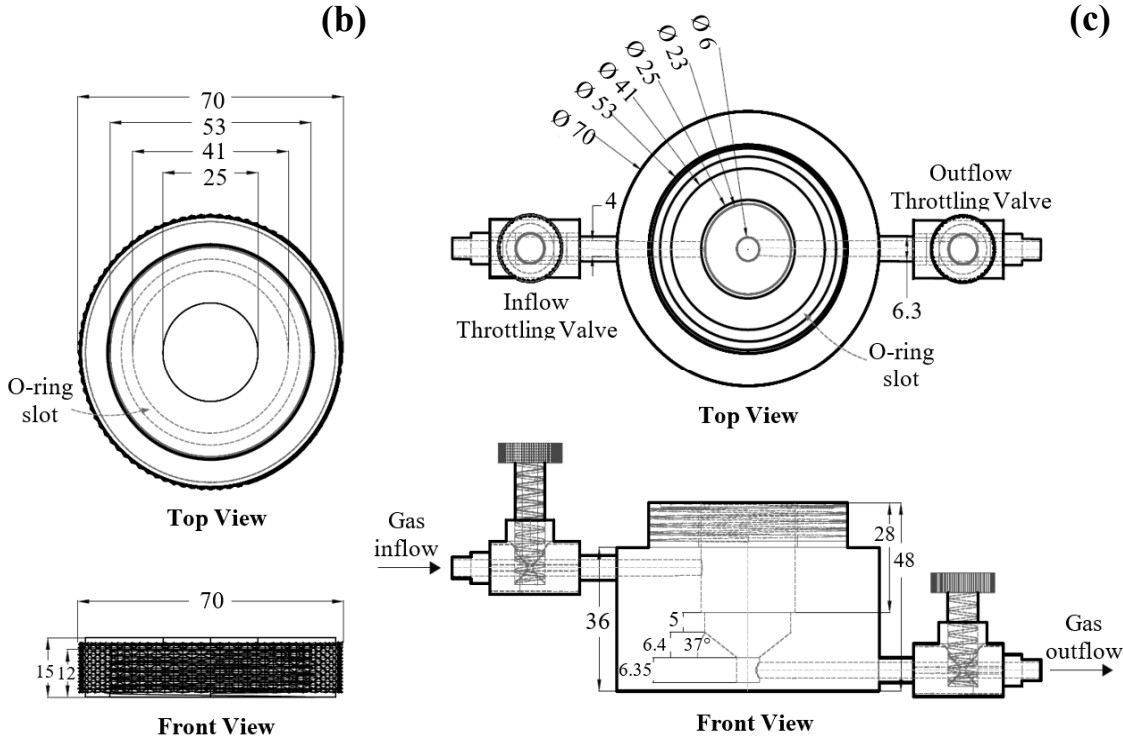

**Figure 6.** Reactor diagram and its blue prints, (**a**) Complete reactor diagram, (**b**) Lid dimensions, and (**c**) Reactor body dimensions.

## 4. Reactor Geometry and Catalyst Support

Even the distribution of light over the surface of the photocatalyst is essential in reaping its full potential. In most of the reactor geometries where light impinges over the surface of the photocatalyst, from center or side, a shadow is casted on the opposite side. Consequently, a major portion of the photocatalyst is not activated. The contact of the light and photocatalyst might be enhanced

by fabricating effective reactor geometries to achieve a uniform distribution of light and better photocatalyst dispersion [6,28,29]. Higher dispersion of the photocatalyst results in enhanced contact with reactants, better mass transfer and guarantees the maximum utilization of the light which all eventually translate to higher quantum yield [30]. To achieve this, a variety of approaches are reported in literature including the utilization of different reactor geometries and catalyst supports, as discussed in detail below.

### 4.1. Monolith Reactor

Tahir et al. studied the effect of photocatalyst dispersion by comparing performance of $TiO_2$ coated micro channel monolith and cell type support (dispersed as single layer over stainless steel cell). Their study revealed a significant enhancement in CO production by $TiO_2$ coated monolith as shown in Figure 7. This increment was mainly attributed to a broader exposed photocatalyst surface available for photocatalytic reaction [6]. In their study with gold-indium $TiO_2$ dispersed over monolith, higher yields were reported particularly formation of $C_2$ and $C_3$ products. This enhancement was linked to effective utilization of photons owing to larger illuminated area of monolith [31]. A similar result of enhancement in $CO_2$ reduction to $CH_4$ was reported for montmorillonite modified $TiO_2$ that was loaded over monolith as compared to the bare one [32].However, their performance is still restricted by limited light penetration despite the high flow rates, minimal pressure drop, and large surface area [33–35].

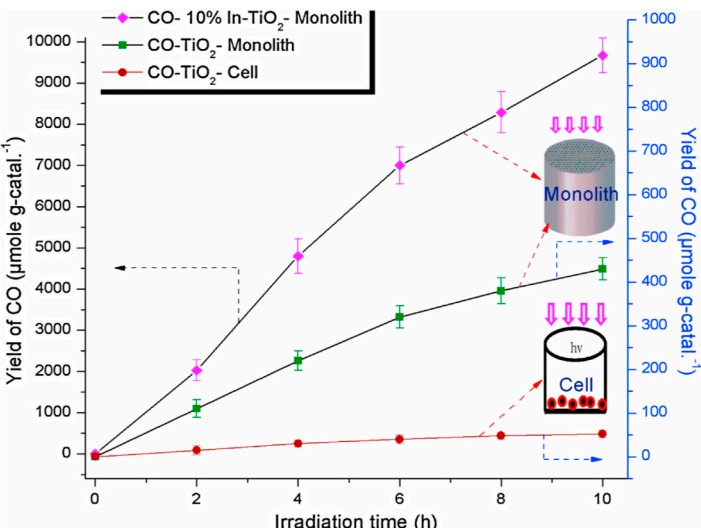

**Figure 7.** Illustration for comparison of $TiO_2$ coated on monolith and cell type support, reproduced with permission from reference [6]. Copyright Elsevier, 2013.

### 4.2. Fiber Optic Reactor

Fiber optic reactors are advantageous when compared to packed bed reactors owing to a better dispersion of photocatalyst and the spreading of light on large surface area [36]. Nguyen et al. compared the yield of the photocatalytic $CO_2$ reduction carried over the photocatalyst coated on optical fiber and glass plate. Their study demonstrated ~15.2 time enhancement in $CH_4$ and 11.6 times in $C_2H_4$ yield, for same amount of photocatalysts. This may be attributed to the synergistic effects of catalyst dispersion and effective light utilization [29]. Wang et al. also carried out $CO_2$ photoreduction while using fiber optic reactor and they attributed the enhancement in yield to the gradual and uniform distribution of light upon irradiation [37]. Although optical fibers accompany the features of catalyst support and effective light distribution, they are saddled with the constraints of limited utilization of reactor volume and the shorter transportation distance of light from the tip of incidence. They occupy 20–30% of reactor volume, but a limited catalyst coated area restricts the effective use of incident light [33,35,37,38].

### 4.3. Monolith Fiber Optic Combined Reactor

Ola et al. combined the mutual effects of higher surface area of monolith and effective light distribution of fiber optics to fabricate internally illuminated monolith reactor, and compared the $CO_2$ reduction performance of this system with slurry reactor. It was found that internal illumination, by optical fibers, of the monolith reactor enhanced quantum efficiency to 23 times owing to the higher surface area of monolith and even the distribution of light by optical fibers [39]. Liou et al. inserted carved polymethylmethacrylate (PMMA) made optical fibers into $NiO/InTaO_4$ coated monolith (honey comb structure) as shown in Figure 8. This reactor when applied for photocatalytic $CO_2$ reduction enhanced the yield of products (methanol and acetaldehyde). An enhancement in the yield can be attributed to increased surface area, higher photocatalyst loading and effective utilization of the light [35].

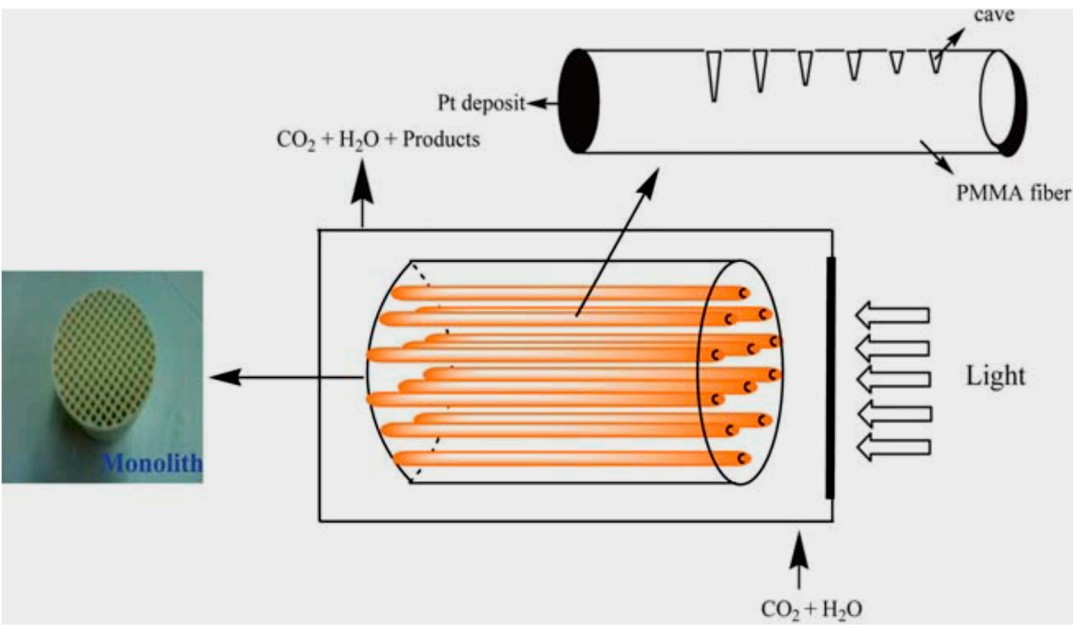

**Figure 8.** Illustration of monolith fiber optic reactor, reproduced from reference [35] with permission from The Royal Society of Chemistry.

## 5. Light Irradiations

Light intensity and the type of irradiations are the most influential parameters for photocatalytic $CO_2$ reduction [40]. Thus, to spur the solar chemical/fuel yielding reactions an effective contact, particularly at the microscopic level, between light and catalyst is imperative. As reported in the literature, a majority of the photocatalysts work efficiently in the UV range thus they are capable of only harnessing a limited range of solar spectrum [41]. To resolve this bottleneck, the modification of semiconductor is carried out for harvesting a wide range of solar spectrum. Other than that, light can be concentrated and channeled to obtain higher photon flux of irradiations [42]. Higher photon flux is not only conducive in propelling apparent quantum efficiency (AQE) but it also shores up the selectivity of multi-electron photoreactions to yield ethane and other long chain solar products. These claims were further vindicated by a study by Nagpal et al. while using $TiO_2$ and CuInS (copper indium sulfide) nanocrystals as the photocatalyst. An enhanced AQY of 4.3% with a higher ethane selectivity of 70% was observed under concentrated sunlight owing to availability of a higher number of electrons [43]. Han et al. performed $CO_2$ photoreduction by concentrating light over $TiO_2$ and $Pt/TiO_2$. The authors performed $CO_2$ photoreduction experiments with different concentrating ratio (CR), which is defined as the ratio of concentrated light flux (amount of energy per unit time per unit area) on the photocatalyst surface to the ambient flux (under non-concentrated conditions). The light irradiation is concentrated and varied by changing the distance between the Fresnel lens (placed in

between light source and photocatalyst) and photocatalyst surface, resulting in different light intensities with different light concentrated focal areas. Their study confirmed that the optimum concentration ratio (CR) significantly increased the AQY by 4.0 and 3.17 times for $TiO_2$ and $Pt/TiO_2$ respectively [42]. Similar results were reported by Li et al. where photocatalytic $CH_4$ yield was improved by 29.5 and 6.2 times under optimum CR for untreated and pre-treated samples, respectively [44]. In their other study, they reported that $CH_4$ yield, for photocatalytic $CO_2$ reduction, over $g-C_3N_4$ at CR10 (10 times concentrating the light) was enhanced by factors of ~11.9 and ~16.0 for untreated and pre-treated samples respectively [45]. Tan et al. also reported enhanced AQY up to optimum light intensity but beyond that, it decreases as the number of photons exceeds the requirement for photocatalytic reaction. They also reported that reaction yields were significantly higher for AM 1.5 filter as compared to the UV cut-off filter which could be attributed to higher photon energy for UV leading to the generation of more photogenerated charges [46]. Based on these studies it can be gauged that better contact of light with photocatalyst under optimally concentrated light may lead to an exceptional yield increase.

## 6. Temperature

Concentrating solar light also increases the temperature depending upon the CR, due to long wavelength irradiations [42,47,48]. Photocatalytic $CO_2$ reduction at an elevated temperatures is promising as it overcomes the thermal barriers that lead to slow reaction rate and marginal yields [48]. The efficacy of temperature rise for photoreaction can be underscored on the basis of enhanced effective collisions among photogenerated charges and reactants that directly relate to the reaction rate [30,46]. Furthermore, elevated temperature also facilitates the desorption of the products providing the way for the adsorption of the $CO_2$ on vacant sites leading to increased reaction rate [49,50]. Wang et al. found that production rate almost doubled when the temperature was increased from 25 °C to 75 °C [37]. Desorption of methanol from the photocatalyst surface increases due to increase in temperature, thus providing more active sites for $CO_2$ photoreduction. The AQY in turn also increased due to more efficient utilization of the incident light and enhanced $CO_2$ adsorption as result of methanol desorption. However, the reaction temperature should not increase too much as it might also desorb the $CO_2$ thus decelerating the photoreduction process. Similarly, Guan et al. attained a temperature up to 590 K for photocatalytic reduction of $CO_2$ and they found temperature rise to be an effective factor for enhancing the yield of solar products [47]. They suggested that such an enhancement was due to the increased collision frequency of photons and diffusion rate of $CO_2$ towards the surface active sites. Hence the thermal energy along with light irradiation can significantly improve the solar products yield by efficiently overcoming the kinetic barrier for $CO_2$ photoreduction reactions. In another study Alxneit et al. revealed the rate of $CH_4$ formation via photocatalytic $CO_2$ reduction increase when the temperature was increased from 25 to 200 °C. Upon further increasing the temperature, the reaction rate decreased due to desorption of the reactants. However, with the aforementioned temperature range, the reaction rate increases with increasing temperature which indicates the importance of thermal step tending to decrease the surface coverage from intermediate species and products. These results clearly suggest that temperature dependence is one of the pivotal factors that influences the photocatalytic reaction rate and thereof the product yield [48]. In another study, Tahir et al. also confirmed the efficacy of temperature rise in photocatalytic $CO_2$ reduction for enhanced CO production over $In/TiO_2$ as shown in the Figure 9 [6]. Such an observation was again explained on the basis of adsorption-desorption phenomenon. Upon increasing the reaction temperature, the mass transfer of $CO_2$ on the active sites is increased, which leads to increased $CO_2$ adsorption, resulting in increased reaction rates. They also observed the raising temperature also transformed the product selectivity towards longer chain hydrocarbons. Similarly Zhang et al. reported, when temperature increased from 323 K to 343 K, the yield of the photocatalytic reaction became twice [50]. They also suggested that enhancement in the product yield is due to desorption of products on the photocatalyst surface, providing more chances of collisions between the excited states and adsorbed reactants. Although

raising temperature is an effective strategy to obtain higher yields, different studies suggest that there is always an optimum temperature range to get the best AQY [42].

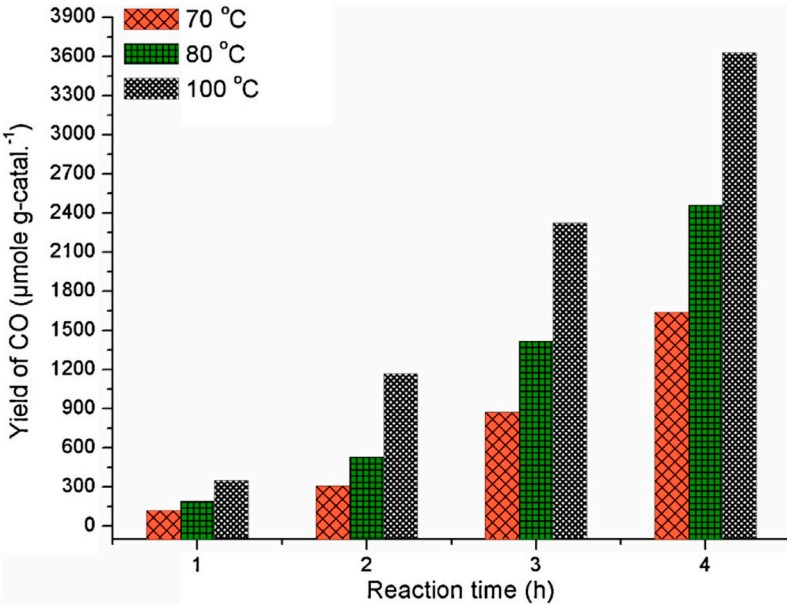

**Figure 9.** Temperature dependent production of CO, reproduced with permission from reference [6]. Copyright Elsevier, 2013.

## 7. Effect of $H_2O/CO_2$ Feed Ratio

An established mixture of water vapors and $CO_2$ gas ($H_2O/CO_2$) is considered to be a cost effective and invulnerable feed for photocatalytic reduction into chemicals/fuels. The feed ratio is another crucial factor with a profound impact over reaction rate and product yield. In addition, the affinity of the photocatalyst for $H_2O/CO_2$ may also lead to well tuning of the product selectivity. Yamashita et al. found highly selective methanol formation over the surface of hydrophilic Ti-Beta(OH) zeolites when compared to conventional Ti-Beta(F) zeolites [51]. Tahir et al. studied the effect of varying $H_2O/CO_2$ ratio by manipulating the $CO_2$ flow rates. In their study they reported that at lower concentration of $CO_2$, water could adsorb over the photocatalyst to react efficiently and give better yields. However, at higher $CO_2$ concentrations, $H_2O$ has to compete for adsorption which may influence the yield. The same authors also studied the variation in $H_2O/CO_2$ ratio and its influence on the yield [6]. Zhang et al. also reported similar results mentioning enhancement in yield with increasing $H_2O/CO_2$ ratio [50]. In another study by Tahir et al., which reports enhancement in $CH_4$ yield with increasing $H_2O/CO_2$ ratio. This is attributed to adsorption of excess water molecules over the photocatalyst and resulting in incremented ability to reduce $CO_2$. Further increase in $CO_2$ decelerated the yield, shown in Figure 10, due to competition between water and $CO_2$ molecules on the photocatalyst active sites [30]. As the $H_2O/CO_2$ feed ratio is increased more water molecules will cover the photocatalyst surface due to hydrophilic nature of the material, which competes with the $CO_2$ molecules to get adsorb on the photocatalyst active sites during the photoreduction process. Therefore, an optimum feed ratio is required for moderate adsorption of both water and $CO_2$ molecules which in turn lead to the maximum $CH_4$ yield. The importance of optimum $H_2O/CO_2$ ratio was further vindicated by Wu et al. in which they reported optimum $H_2O/CO_2$ ratio is essential for enhanced yield of methanol [52]. The investigation of Tan et al. also stressed the need for adsorption of optimum number of molecules of $CO_2$ and $H_2O$ over the catalyst surface to obtain enhanced yields. Moreover it is important to note that, enhanced adsorption of any reactant will certainly hinder the adsorption of others [46]. Thus, a tradeoff should exist between $CO_2$ and $H_2O$ to mitigate their competitive adsorption over the surface

of the photocatalyst. Thus, due to rivalry in adsorption, an optimum feed ratio is indispensable to get improved yield.

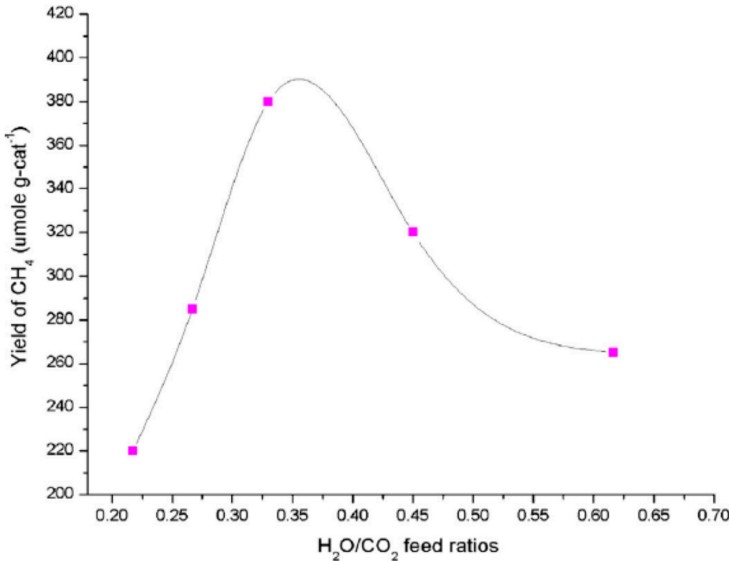

**Figure 10.** Effect of $H_2O/CO_2$ ratio over $CH_4$ yield, reproduced with permission from reference [30]. Copyright Elsevier, 2013.

## 8. Other Factors

In addition to the above mentioned parameters, there exists several other parameters which also alters the photocatalytic yield. A reported work investigated the performance of photocatalyst in cell type and monolith photoreactor which revealed that broader exposed area of photocatalyst, as for monolith catalyst support, improves the yield (shown in Figure 11). This can be related to enhancement in the illumination area of photocatalyst i.e., exposure of same weight of the catalyst in a larger area [29,30,39,42,53]. In addition to the conditions of high purity, ultra-high vacuum also ensures the absence of external impurities [9]. Moreover, by improving the surface properties like $CO_2$ adsorption capability, photocatalytic yield can also be improved [44,45,54–58].

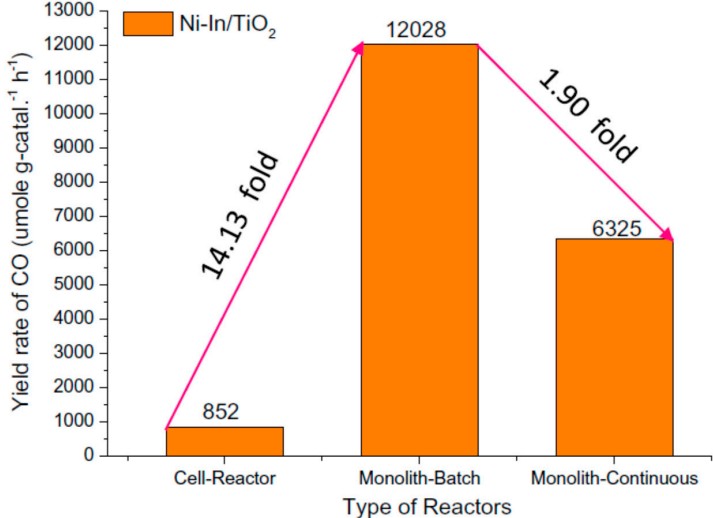

**Figure 11.** Performance evaluation of cell type and monolith batch/continuous photoreactor for $CO_2$ photoreduction, reproduced with permission from reference [53]. Copyright Elsevier, 2015.

## 9. Benchmarking for Performance Evaluation

Based on the above discussion it can be well acknowledged that reactor geometries, reaction parameters, light intensity and wavelength are important parameters that play a pivotal role to control the product yields and selectivity. Thus photocatalysts that were tested under different conditions cannot be compared on equal grounds [42]. Like, Tan et al. reported that a maximum yield of 3.14 $\mu$mol g$^{-1}$ was obtained at a light intensity of 177.2 mW cm$^{-2}$ but in terms of AQY, the performance is better at 81 mW cm$^{-2}$, as shown in Figure 12, for a given amount of photocatalyst [46]. This clearly reveals that although the photocatalyst is producing maximum yield at 177.2 mW cm$^{-2}$ it still underperforms in capitalizing photogenerated charges. Thus, reporting performance of the photocatalytic reaction only on the basis of intrinsic capability of photocatalyst to yield the products is not a rationalized approach. On the contrary, evaluating the performance of the photocatalyst in terms of AQY, which incorporates reactor area, incident and harvested light, can be a more adequate approach [59]. For meaningful performance comparison, AQYs of the different photocatalysts are calculated and presented in Tables 1–3, using Equations (4)–(8) [11,59].

$$\text{AQY (\%)} = \frac{\text{number of reacted electrons}}{\text{effective number of incident photons}} \times 100\%, \tag{4}$$

$$\text{number of reacted electrons} = \left[\begin{array}{c} \text{mole of product} \\ \text{produced in time, t} \end{array}\right] \times \left[\begin{array}{c} \text{number of electrons} \\ \text{required to produce} \\ \text{1 mol of product} \end{array}\right] \times N_A, \tag{5}$$

$$\text{effective number of incident photons} = \frac{\text{light absorbed by the photocatalyst}}{\text{average photon energy}} \times t, \tag{6}$$

$$\text{light absorbed by the photocatalyst} = H \times A, \tag{7}$$

$$\text{average photon energy} = \frac{hc}{\lambda}, \tag{8}$$

where H is the apparent light input (W m$^{-2}$), A is the geometric irradiation area (m$^2$), h is Planck's constant ($6.626 \times 10^{-34}$ J·s), c is speed of light ($3 \times 10^8$ m s$^{-1}$), $\lambda$ is the average wavelength of light source (nm) and N$_A$ is Avogadro's number ($6.022 \times 10^{23}$ atoms mol$^{-1}$).

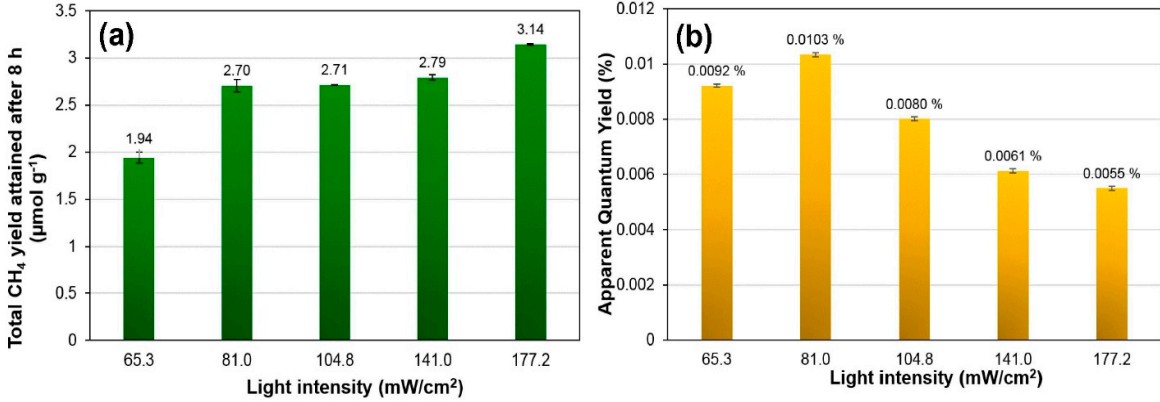

**Figure 12.** Effect of different light intensities at (**a**) CH$_4$ yield and (**b**) apparent quantum yield(AQY) at variable light intensities, reproduced with permission from reference [46]. Copyright Elsevier, 2016.

**Table 2.** Summary of operating parameters and yields used by other research works for photocatalytic $CO_2$ reduction using Batch reactor.

| Photocatalyst | Synthesis Method | Pre-Treatment of Reactor | Light Source | Reducing Agent | Reaction Parameters | Reactor | Main Product | AQY (%) |
|---|---|---|---|---|---|---|---|---|
| $Bi_2WO_6$/Au/CdS Z-scheme [61] | bath deposition method | vacuum-treated several times, and then filled with high purity $CO_2$ gas | 300 W Xe lamp ($\lambda$ > 400 nm) | 0.4 mL of DI water | 100 mg catalyst; 230 mL reactor; 1 mL sample gas; ambient pressure; 8 h irradiation | Batch reactor | $CH_4$ @ ~0.75 $\mu$mol g$^{-1}$ h$^{-1}$ | 0.012 [b] |
| rGO-CuO hybrid structure [62] | covalent grafting | purged with nitrogen gas for 15 min then purged with $CO_2$ for 30 min under continuous stirring | 20 W white cold LED flood light (200 < $\lambda$ < 700 nm); 85 W m$^{-2}$ | mixture of DMF (45 mL) and $H_2O$ (5 mL) | 100 mg catalyst; 100 mL reactor; 20 $\mu$L total sample gas analyzed; 24 h irradiation | Batch reactor | $CH_3OH$ @ 1282 $\mu$mol g$^{-1}$ | 0.013 [b,c] |
| $Cd_{1-x}Zn_xS$ solid solution [63] | two-step self-templated synthesis | purged with argon for 1 h, then 2 mL of deionized water was injected, and purged with ultra-pure $CO_2$ for 30 min | 100 W LED plate with collimating lens; visible light ($\lambda$ = 450 nm); 285 mW cm$^{-2}$ | 2 mL of DI water | 45 mg catalyst; 130 cm$^3$ reactor; 250 $\mu$L sample gas extracted every 1 h; 1 atm; 25 °C; 5 h irradiation | Batch reactor | CO @ 2.90 $\mu$mol g$^{-1}$ h$^{-1}$ (AQY$_{CO}$ = 0.016) $CH_4$ @ 0.22 $\mu$mol g$^{-1}$ h$^{-1}$ (AQY$_{CH4}$ = 0.005) | 0.02 [a] |
| Co-ZIF-9/$TiO_2$ nanostructure [64] | in situ growth method | purged with high-purity $CO_2$ gas | 300 W Xe lamp (200 < $\lambda$ < 900 nm); 494 mW cm$^{-2}$ | 3 mL DI water | 50 mg catalyst; 390 mL reactor; 0.5 mL sample gas extracted; 70 kPa; 10 h irradiation | Batch reactor with gas circulation system | CO @ 17.58 $\mu$mol g$^{-1}$ h$^{-1}$ | 0.053 [a] |
| Pt/$TiO_2$ mesoporous structure [65] | soft-template method | purged with high purity $CO_2$ bubbled through DI water for more than 1 h | 350 W Xe-lamp with 420 nm cutoff filter; UV light @ 34.8 mW cm$^{-2}$ | $H_2O$ | 100 mg catalyst; 159 mL tubular reactor (length: 28 cm, Ø: 3 cm); 60 ± 2 °C; 2 h irradiation | Batch reactor | $CH_4$ @ 5.7 $\mu$mol g$^{-1}$ | 0.064 [a] |
| $In_2O_3$–$C_3N_4$ hybrid structure [66] | simple solvothermal method | purged with high-purity $CO_2$ gas | 500 W Xenon lamp; 1200 mW cm$^{-2}$ | 0.1 mL ultrapure $H_2O$ | 20 mg catalyst; 90 mL reactor; 4 h irradiation | Batch reactor | $CH_4$ @ 159.2 ppm | 0.082 [a] |
| Pd/(10 wt.% LDH/$C_3N_4$) hybrid structure [67] | electrostatic interaction | introduction of 200 torr $CO_2$ into the system | 500 W Hg (Xe) lamp without filter | 100 mL $H_2O$ | 200 mg catalyst; 200 $\mu$L sample gas extracted; AQY @ $\lambda$ = 420 nm; 200 torr; 72 h irradiation | Batch reactor | $CH_4$ @ 6.5 $\mu$mol | 0.093 [b] |
| In/$TiO_2$-monolith [6] | sol–gel single step method | continuous passing of $CO_2$, He and $H_2O$ mixture through the reactor for about 1 h | 200 W Hg lamp for UV irradiations ($\lambda$ < 252 nm); 150 mW cm$^{-2}$ | $H_2O$ | 50 mg catalyst; 150 cm$^3$ reactor; 1000 $\mu$L sample gas extracted; $P_{CO2}$ = 0.20 bar; $P_{H2O}$ = 0.074 bar; 10 h irradiation | Batch reactor | CO @ 962 $\mu$mol g$^{-1}$ h$^{-1}$ | 0.10 [b] |
| $TiO_{2-x}$/$CoO_x$ hybrid structure [68] | (own method) | blown with $CO_2$ for 20 min | 150 W UV lamp; 20 mW cm$^{-2}$ | 2 mL of DI water | 50 mg catalyst; 100 mL reactor; 1.01 bar; room temperature; 4 h irradiation | Batch reactor | CO @ 1.247 $\mu$mol g$^{-1}$ h$^{-1}$ (AQY$_{CO}$ = 0.0817) $CH_4$ @ 0.0903 $\mu$mol g$^{-1}$ h$^{-1}$ (AQY$_{CH4}$ = 0.0237) | 0.105 [a] |

**Table 2.** *Cont.*

| Photocatalyst | Synthesis Method | Pre-Treatment of Reactor | Light Source | Reducing Agent | Reaction Parameters | Reactor | Main Product | AQY (%) |
|---|---|---|---|---|---|---|---|---|
| Ag-Au/TiO$_2$ nanowires [69] | facile hydrothermal synthesis | compressed CO$_2$ and H$_2$ were continuously passed through the reactor | 35 W HID Xe lamp; 20 mW cm$^{-2}$ | H$_2$ | 10 mg catalyst; 108 cm$^3$ reactor; 0.20 bar; 4 h irradiation | Batch reactor | CO @ 1813 μmol g$^{-1}$ h$^{-1}$ | 0.1108 [b] |
| LaPO$_4$–Pt nanorods [70] | hydrothermal method and photo deposition | reactor was evacuated and filled with CO$_2$ for 1 h with stirring | 125 W high-pressure Hg lamp (λ < 365 nm) | 70 mL H$_2$O | 50 mg catalyst; 200 mL reactor; 1 atm; 20 °C; 4 h irradiation | Batch reactor | CH$_4$ @ 0.62 μmol g$^{-1}$ | 0.15 [b] |
| Zn$_2$GeO$_4$ micro/mesoporous [71] | simple ion exchange | vacuum-pumped and washed with high purity CO$_2$ gas | 300 W Xe arc lamp (λ = 251 ± 16 nm) | 0.5 mL DI water | 200 mg catalyst; 360 mL reactor; 0.5 mL sample gas extracted; ambient pressure; 12 h irradiation | Batch reactor | CH$_4$ @ 9.5 ppm g$^{-1}$ h$^{-1}$ | 0.20 [b] |
| ZnIn$_2$S$_4$ one-unit-cell atomic layers [72] | (own method) | vacuum-treated three times, then pumped with high-purity CO$_2$ | PLS-SXE300/ 300 UV Xe lamp; 100 mW cm$^{-2}$ | 2 mL DI water | 100 mg catalyst; atmospheric pressure; 298 ± 0.2 K; 1 h irradiation | Batch reactor | CO @ 33.2 μmol g$^{-1}$ h$^{-1}$ | 0.23 [b] |
| Cu$_2$O/x% RGO composites [73] | microwave-assisted hydrothermal reaction | CO$_2$ purged | 150 W Xe lamp; 540 μW cm$^{-2}$ | 3 mL DI water | 500 mg catalyst; 120 mL reactor; sample gas extracted every 30 min; 20 h irradiation | Batch reactor | CO @ 50 ppm g$^{-1}$ h$^{-1}$ | 0.34 [b] |
| Pt/MgAl-LDO/TiO$_2$ hybrid structure [74] | in-situ deposition, calcination and photo deposition | degassed for 30 min, and then bubbled with CO$_2$ till the pressure reaches 1 atm | 300 W Xe lamp; 1.1 mW cm$^2$ | H$_2$O | 20 mg catalyst; AQY @ λ = 365 nm; 1 atm; 20 °C; 8 h irradiation | Batch reactor | CH$_4$ @ 0.11 μmol | 0.35 [b,c] |
| LDH/RGO/CN hybrid structure [75] | hydrothermal synthesis and in situ loading | vacuum-treated several times, and then flowed with high purity CO$_2$ gas | 300 W Xe arc lamp; 1.8 mW cm$^{-2}$ | 4 mL DI water | 50 mg catalyst; 420 mL reactor; 1 mL sample gas extracted; AQY @ λ = 385 nm; ambient pressure; 5 h irradiation | Batch reactor | CO @ 10.11 μmol g$^{-1}$ h$^{-1}$ | 0.45 [b] |
| Cu$_2$O/WO$_3$ nanosheets [76] | modified method | vacuum treated, and then purged several times with high purity CO$_2$ gas | 300 W Xenon arc lamp with a UV cutoff filter (λ > 400 nm) | H$_2$O | 85 mg catalyst; 18 h irradiation | Batch reactor | CO @ 0.56 μmol g$^{-1}$ h$^{-1}$ | 0.503 [b] |
| TiO$_2$ microsphere [77] | sol–gel approach | introduction of pressurized CO$_2$ @ (50 psi) | 40 W Hg UV lamp (λ = 254 nm); 20 mW cm$^{-2}$ | 100 μL H$_2$O | 200 mg catalyst; 39 mm diameter and 9 mm depth reactor; 10 mL sample gas extracted; 50 psi; 24 h irradiation | Batch reactor | CO @ 0.56 μmol g$^{-1}$ h$^{-1}$ (AQY$_{CO}$ = 0.204) CH$_4$ @ 0.94 μmol g$^{-1}$ h$^{-1}$ (AQY$_{CH4}$ = 0.34) | 0.54 [b,c] |

**Table 2.** *Cont.*

| Photocatalyst | Synthesis Method | Pre-Treatment of Reactor | Light Source | Reducing Agent | Reaction Parameters | Reactor | Main Product | AQY (%) |
|---|---|---|---|---|---|---|---|---|
| RGO-CdS nanorod composites [78] | microwave hydrothermal route | degassed with nitrogen for 30 min | 300 W Xe arc lamp with a UV-cutoff filter ($\lambda \geq$ 420 nm); 150 mW cm$^{-2}$ | 10 mL distilled water | 100 mg catalyst; 200 mL reactor; 1 mL sample gas extracted every 1 h; atmospheric pressure and ambient temperature; 3 h irradiation | Batch reactor | CH$_4$ @ 2.51 µmol g$^{-1}$ h$^{-1}$ | 0.80 [b] |
| HCP-TiO$_2$-FG composite [79] | in situ growth | - | 300 W Xe lamp ($\lambda \geq$ 420 nm); 433 mW cm$^{-2}$ | H$_2$O | 20 mg catalyst; standard atmospheric pressure; 5 h irradiation | Batch reactor | CH$_4$ @ 27.62 µmol g$^{-1}$ h$^{-1}$ (AQY$_{CH4}$ = 1.14) CO @ 21.63 µmol g$^{-1}$ h$^{-1}$ (AQY$_{CO}$ = 0.2227) | 1.36 [a] |
| Co/Pal heterostructure [80] | in situ electrostatic adsorption deposition process | filled with high purity CO$_2$ gas | 300 W Xe lamp | 5 mL acetonitrile/H$_2$O (4:1) | 9 mg photosensitizer + 1 mg co-catalyst + 1 mL TEOA; 80 mL reactor; AQY @ $\lambda$ = 420 nm; 1 atm; 25 °C; 6 h irradiation | Batch reactor | CO @ ~86 µmol | 1.38 [b] |
| CuO−TiO$_2$ hollow microspheres [81] | one-pot template-free synthesis | introduction of pressurized CO$_2$ @ (50 psi) | 40 W Hg UV lamp($\lambda$ = 254 nm); 20 mW cm$^{-2}$ | 200 µL H$_2$O | 10 mg catalyst; reactor diameter of 39 mm and a depth of 9 mm; 50 psi; 24 h irradiation | Batch reactor | CO @ 14.5 µmol g$^{-1}$ h$^{-1}$ (AQY$_{CO}$ = 1.285) CH$_4$ @ 2.1 µmol g$^{-1}$ h$^{-1}$ (AQY$_{CH4}$ = 0.747) | 2.03 [b] |
| Pt-TiO$_2$ spheres [77] | microwave-assisted solvothermal method | introduction of pressurized CO$_2$ @ (50 psi) | 40 W Hg UV lamp ($\lambda$ = 254 nm); 20 mW cm$^{-2}$ | 100 µL H$_2$O | 200 mg catalyst; 39 mm diameter and 9 mm depth reactor; 10 mL sample gas extracted; 50 psi; 24 h irradiation | Batch reactor | CO @ 18.9 µmol g$^{-1}$ h$^{-1}$ (AQY$_{CO}$ = 1.632) CH$_4$ @ 3.6 µmol g$^{-1}$ h$^{-1}$ (AQY$_{CH4}$ = 1.315) | 2.95 [b,c] |
| Pd$_x$Cu$_1$-TiO$_2$ hybrid structures [82] | in situ growth | filled with 0.2 MPa CO$_2$ for 60 min | 300 W Xe lamp ($\lambda$ < 400 nm); 2 mW cm$^{-2}$ | H$_2$O | 5 mg of TiO$_2$ + 0.01 mmol of metal atoms for catalyst; 100 mL reactor; 0.2 MPa; 2 h irradiation | Batch reactor | CH$_4$ @ 19.6 µmol g$^{-1}$ h$^{-1}$ | 12.53 [a] |
| In/TiO$_2$ nanoparticles [60] | sol–gel single step method | purged with CO$_2$ and He for an hour | 500 W mercury flash lamp ($\lambda$ = 365 nm); 40 mW cm$^{-2}$ | H$_2$O | 0.25 mg catalyst; 106 cm$^3$ reactor; 1000 µL sample gas extracted; 0.20 bars, 373 K; 8 h irradiation | Batch reactor | CH$_4$ @ 244 µmol g$^{-1}$ h$^{-1}$ (AQY$_{CH4}$ = 42.39) CO @ 81 µmol g$^{-1}$ h$^{-1}$ (AQYCO = 3.52) | 45.91 [a] |

**Table 2.** *Cont.*

| Photocatalyst | Synthesis Method | Pre-Treatment of Reactor | Light Source | Reducing Agent | Reaction Parameters | Reactor | Main Product | AQY (%) |
|---|---|---|---|---|---|---|---|---|
| $ZnV_2O_4$ microspheres [83] | one-step hydrothermal process | purged with $CO_2$ gas carrying $H_2O$ for 30 min | 35 W HID Xe lamp; 20 mW cm$^{-2}$ | $H_2O$ | 100 mg catalyst; $CO_2$ flowrate @ 20 mL min$^{-1}$; 108 cm$^3$ reactor; 0.20 bar; 100 °C; 4 h irradiation | Batch reactor | CO @ 485 µmol g$^{-1}$ h$^{-1}$ (AQY$_{CO}$ = 31.92) $CH_3OH$ @ 100 µmol g$^{-1}$ h$^{-1}$ (AQY$_{CH3OH}$ = 19.75) | 51.67 [a] |
| $NiO/InTaO_4$ monolith coated structure [35] | impregnation method and sol-gel method | purged overnight using a flow of He then switched to pure $CO_2$ with saturated water vapor for 1 h | 300 W Xe arc lamp with AM 1.5 filter; 100 mW cm$^{-2}$ | $H_2O$ | 88.7 mg catalyst; 216 cm$^3$ reactor; 1 bar; 70 °C; 2 h irradiation | - | $CH_3OH$ @ 0.16 µmol g$^{-1}$ h$^{-1}$ (AQY$_{CH3OH}$ = 0.012) $CH_3CHO$ @ 0.3 µmol g$^{-1}$ h$^{-1}$ (AQY$_{CH3CHO}$ = 0.058) | 0.07 [b] |
| MAT nanofibers [84] | (own method) | blown with nitrogen for 30 min | 300 W simulated solar Xe arc lamp | $H_2O$ | 200 mL reactor; 1 mL sample gas extracted every 1 h; atmospheric pressure and ambient temperature; 3 h irradiation | - | $CH_4$ @ 0.86 µmol g$^{-1}$ h$^{-1}$ | 0.091 [b] |
| BiOI few-layered nanosheets [85] | (own method) | thoroughly vacuum-treated | 300 W high pressure Xe lamp | 5 mL $H_2SO_4$ & 1.712 g $NaHCO_3$ | 150 mg catalyst; 500 mL reactor; 0.15 mL sample gas extracted; 20 °C AQY @ λ = 420 nm; 4 h irradiation | - | CO @ 0.615 µmol h$^{-1}$ $CH_4$ @ 0.063 µmol h$^{-1}$ | 0.140 [b] |
| $CdS–WO_3$ heterostructure [86] | simple precipitation method | blown with nitrogen for 30 min | 300 W Xe arc lamp with a UV-cutoff filter (λ ≥ 420 nm); 6.0 mW cm$^{-2}$ | $H_2O$ | 100 mg catalyst +10 mL of distilled water to form films; 200 mL reactor; 1 mL sample gas extracted every 1 h; AQY @ λ = 420 nm; atmospheric pressure and ambient temperature | - | $CH_4$ @ 1.02 µmol g$^{-1}$ h$^{-1}$ | 0.40 [b] |
| $CeO_x$-S/$ZnIn_2S_4$ hybrid structure [87] | one-pot hydrothermal method | introduction of high purity $CO_2$ gas into the reactor for 3 min | 9.0 W (455 nm LEDs) | 0.5 mL $H_2O$ | 10 mg catalyst; 6.98 mL reactor; 1 bar; below 42 °C; 10 h irradiation | - | CO @ 0.18 mmol g$^{-1}$ h$^{-1}$ | 1.34 [b] |
| $Pt/TiO_2$ [42] | vacuum impregnation | reactor was cleaned with nitrogen for half an hour then it was replaced and saturated with $CO_2$ gas for at least 30 min | 300 W UV light; 10 mW cm$^{-2}$ | 2 mL $H_2O$ | 100 mL reactor; sample gas analyzed every 1 h; 0.1 MPa; 7 h irradiation | - | $CH_4$ @ 20.55 µmol g$^{-1}$ | 10.03 [b] |

a = AQY computed; b = AQY was computed by authors of the reference paper; c = AQY was computed by authors of the reference paper multiplying it with mass of photocatalyst.

Table 3. Summary of operating parameters used by other research works for photocatalytic $CO_2$ reduction using Continuous flow reactor.

| Photocatalyst | Synthesis Method | Pre-Treatment of Reactor | Light Source | Reducing Agent | Reaction Parameters | Reactor | Main Product | AQY (%) |
|---|---|---|---|---|---|---|---|---|
| TiO$_2$/NRGO-300 nanocomposites [88] | one-step urea-assisted hydrothermal method | purged with $CO_2$ at 16 mL min$^{-1}$ for 40 min | 400 W Xe lamp (250 < λ < 400 nm); 11.5 mW cm$^{-2}$ | $H_2O$ | 10 mg catalyst; $CO_2$ flowrate @ 3 mL min$^{-1}$; sample gas extracted every 1 h; 8 h irradiation | Continuous flow reactor | CO @ 356.5 μmol g$^{-1}$ | 0.0072 [b,c] |
| 5GO–OTiO$_2$ (UV light) hybrid heterostructure [46] | facile wet chemical impregnation technique | purged with wet $CO_2$ at 30 mL min$^{-1}$ for 30 min | 500 W Xe arc lamp with a UV filter (λ > 400 nm); 81.0 mW cm$^{-2}$ | $H_2O$ | $CO_2$ flowrate @ 5 mL min$^{-1}$; Quartz column reactor (ID = 9 mm, OD = 11 mm, length = 250 mm); sample gas extracted every 0.5 h; 1 bar; 25 ± 5 °C; 8 h irradiation | Continuous flow reactor | CH$_4$ @ 2.7 μmol g$^{-1}$ | 0.0103 [b] |
| TiO$_2$ nanofibers [89] | sol-gel method and electrospinning technique | firstly, degassed under vacuum and then purged with Ar for 1 h, then fed with $CO_2$/$H_2O$ mixture in dark for 1 h, then reactor was pressurized and kept at a reaction flow rate of 2 mL min$^{-1}$ for another 1 h. | four 6 W UV lamps (λ$_{max}$ = 365 nm) | $H_2O$ | 100 mg catalyst; 190 mL reactor; 7.25 $CO_2$:$H_2O$ molar ratio; sample gas analyzed every 22 min; 2 bars; 50 °C; 20 h irradiation | Continuous flow reactor | CO @ 203.91 μmol g$_{cat}$$^{-1}$ | 0.04 [b] |
| Cu/GO-2 hybrid structure [90] | one-pot microwave process | purged with nitrogen gas for 1 h then followed by pure $CO_2$ for another 1 h | 300 W halogen lamp; 100 mW cm$^{-2}$ | $H_2O$ | 100 mg catalyst; 300 mL reactor; $CO_2$ flowrate @ 4 μL/min; 25.0 ± 0.5 °C; 2 h irradiation | Continuous flow reactor | CH$_3$OH @ 2.94 μmol g$^{-1}$ h$^{-1}$ (AQY$_{CH3OH}$ = 0.0296) CH$_3$CHO @ 3.88 μmol g$^{-1}$ h$^{-1}$ (AQY$_{CH3CHO}$ = 0.065) | 0.095 [a] |
| G/TiO$_2$-001/101 nanocomposites [91] | one-pot solvothermal method | purged with the $CO_2$ + $H_2O$ mixture at 200 mL min$^{-1}$ for 1 h and then reduced to 5 mL min$^{-1}$ for 30 min | 300W Xe arc lamp (300 < λ < 400 nm); 20.5 mW cm$^{-2}$ | 5 mL DI water | 10 mg catalyst; 85 mL reactor; sample gas analyzed every 30 min; atmospheric pressure; 120 °C; 4 h irradiation | Continuous flow reactor | CO @ 70.8 μmol g$^{-1}$ h$^{-1}$ | 0.0864 [b,c] |
| BWO-OV/BiOI binanosheets [92] | simple self-assembly approach | purged with the $CO_2$/$H_2O$ gas mixture at 50 mL min$^{-1}$ for 30 min | 500 W Xenon arc lamp with UV cut-off filter (to remove λ < 400 nm) | $H_2O$ | $CO_2$ flowrate @ 5 mL min$^{-1}$; sample gas analyzed every 1 h; atmospheric pressure and ambient temperature; 8 h irradiation | Continuous flow reactor | CO @ 320.19 μmol g$^{-1}$ CH$_4$ @ 18.32 μmol g$^{-1}$ | 0.432 [b] |
| Pt$^{2+}$–Pt$^0$/TiO$_2$ nanoparticles [93] | sol–gel method | purged with $CO_2$ + $H_2O$ mixture at 200 mL min$^{-1}$ for 1 h and then at 3 mL min$^{-1}$ for another 30 min. | 300 W Xe arc lamp UV light irradiation (320 < λ < 420 nm); 32.5 mW cm$^{-2}$ | $H_2O$ | 200 mg catalyst; 85 mL reactor; sample gas extracted every 40 min; 50 °C; 7 h irradiation | Continuous flow reactor | CH$_4$ @ 264.5 μmol g$^{-1}$ (AQY$_{CH4}$ = 1.35) CO (AQY$_{CO}$ = 0.07) | 1.42 [b] |
| (Pt/TiO$_2$) @rGO core-shell-structured [94] | hydrothermal method | vacuum-treated, then purged with $CO_2$ gas @ 50 cm$^3$ min$^{-1}$ for 30 min | 300 W Xe lamp (320 < λ < 780 nm); 80 mW cm$^{-2}$ | 2.0 mL $H_2O$ | 100 mg catalyst; sample gas extracted every 1 h; 0.1 MPa; 4 °C; 8 h irradiation | Continuous flow reactor | CH$_4$ @ 41.3 μmol g$^{-1}$ h$^{-1}$ | 1.93 [b,c] |

**Table 3.** *Cont.*

| Photocatalyst | Synthesis Method | Pre-Treatment of Reactor | Light Source | Reducing Agent | Reaction Parameters | Reactor | Main Product | AQY (%) |
|---|---|---|---|---|---|---|---|---|
| NiO/Ni-GR nanoparticles [95] | pyrolysis and incipient wetness impregnation | photoreactor was heated at different temperatures | 300 W Xe lamp; 2236 W m$^{-2}$ | H$_2$ | 40 mg catalyst; 51 mL reactor; 1.3 bar; 200 °C; 2 h irradiation | Continuous flow reactor | CH$_4$ @ 642 μmol g$_{Ni}$$^{-1}$ h$^{-1}$ | 1.98 [b] |
| Pt-TiO$_2$ nanostructured films [96] | aerosol chemical vapor deposition | purged with CO$_2$ and water vapor at 100 mL min$^{-1}$ for 1 h, and then reduced and maintained at 3 mL min$^{-1}$ | 400 W Xe lamp (250 < λ < 388 nm); 19.6 mW cm$^{-2}$ | H$_2$O | 0.7 mg catalyst; atmospheric pressure and room temperature; 5 h irradiation | Continuous flow reactor | CH$_4$ @ 1361 μmol g$^{-1}$ h$^{-1}$ (AQY$_{CH4}$ = 2.33) CO @ 179.34 μmol g$^{-1}$ h$^{-1}$ (AQY$_{CO}$ = 0.077) | 2.41 [b,c] |

a = AQY computed; b = AQY was computed by authors of the reference paper; c = AQY was computed by authors of the reference paper multiplying it with mass of photocatalyst.

Values for the calculations of AQYs included in this review are given in detail in Supplementary Tables S1 and S2. Moreover, prior to photocatalytic testing under $CO_2$, repeated photocatalytic reduction tests with $H_2O$ and inert gas can rule out the possible role of organic contamination. Further confirmation of the carbon source of photocatalytic products can be ascertained by an isotopic labeling test while using $^{13}CO_2$. Additionally, optimization of the reaction parameters, condition of high purity and effective photocatalyst and light contact can provide an even better judgment of the performance [1,4,9,60].

## 10. Conclusions

Photocatalytic $CO_2$ reduction is a fascinating approach owing to its two-fold benefits of $CO_2$ abatement and its subsequent conversion to renewable fuels/chemicals. However, optimizing this process has been a challenge for researchers as a variety of process parameters and reaction conditions indistinctly influence the product yield. One parameter is the reaction temperature which by increasing overcomes the kinetic barrier; but on the contrary, excessive temperature might also lead to decomposition. The reactants feed ratio also affects the product yield: at an optimum value, it synergizes the product yield and selectivity but beyond which, the non-stoichiometric ratios of $H_2O/CO_2$ are competing to get adsorbed on the photocatalyst, resulting in decreased productivity. In addition, optimal light intensity is imperative to produce stoichiometric amount of photogenerated charges for efficient transformation of adsorbed $H_2O/CO_2$ to solar fuel/chemicals; surplus of these charges, produced at higher light intensities, will recombine and eventually reduce the AQY. Moreover, productivity enhancement also depends on the choice of reactor type: continuous flow reactors have shown potential in overcoming the low yield, re-adsorption, and decomposition of photocatalytic products accustomed to batch reactors. Additionally, the reactor geometry with better light and photocatalyst contact notably affects the production yield. Knowing how these parameters affect the product yield, performance of the photocatalysts tested under different conditions are inappropriate to compare in frequently used customary units i.e., $\mu$mol g$^{-1}$ or ppm cm$^{-2}$. Hence, with the consideration of various process parameters and reaction conditions i.e., reaction temperature, feed ratio, irradiation source, reactor type and geometry, reporting yield of the photocatalysts in terms of AQYs rather than production rate is a more appropriate and pragmatic approach. Additionally, the removal of organic contamination, isotopic labelling, and photocatalytic $CO_2$ reduction under inert and ultra-sealed condition can provide more realistic assessment of performance.

The overall aim of the present review article is to highlight the influence of key process and reaction parameters on $CO_2$ photoreduction process to be used for benchmarking. While considering all the aforementioned parameters, a continuous and ultra-sealed gas phase reactor purged under high vacuum with an optimum blend of temperature, feed ratio, and offering larger contact between light and photocatalyst can be an effective approach for achieving maximum yield with the effective utilization of incident photons. Moreover, in order to compare the photocatalytic system performance, a unified parameter should be recognized; reporting yield in terms of AQY standardizes the photocatalytic performance analysis by normalizing the effect of the most influential parameters. Our aforementioned photoreaction setup and studies, well matching with the concluded benchmarking criteria, can be followed for future studies in the field of gas phase $CO_2$ photoreduction.

**Supplementary Materials:** The following are available online at http://www.mdpi.com/2073-4344/9/9/727/s1. Table S1: Computation of AQY performance of photocatalysts by In et al., Table S2: Computation of AQY performance of photocatalysts from other research works.

**Author Contributions:** S.-I.I. conceptualized and edited the manuscript, S.A., M.C.F., A.R., S.S., C.B.H., H.R.K., Y.H.P., Y.H., H.S.K., H.K., E.H.G., J.L. and D.K. wrote the manuscript.

**Funding:** The authors gratefully acknowledge the support of the Ministry of Science and ICT (2017R1E1A1A01074890 and 2017M2A2A6A01070912). This research was also supported by the Technology Development Program to Solve Climate Changes of the National Research Foundation (NRF) funded by the Ministry of Science and ICT (2015M1A2A2074670) as well as by the DGIST R&D Program of the Ministry of Science and ICT (19-BD-0404) and supported by a grant of the Korea Health Technology R&D Project through the Korea Health Industry Development

Institute (KHIDI), funded by the Ministry of Health & Welfare, Republic of Korea (HI19C0506) and supported by Flux Photon Corporation.

**Conflicts of Interest:** The authors declare no conflict of interest.

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
