# Peer review of "Gas Phase Photocatalytic CO2 Reduction, “A Brief Overview for Benchmarking”"

_catalysts, doi:10.3390/catal9090727_

Round 1

Reviewer 1 Report

Increasing concentration of CO2 from human activities is believed to be the reason for global warming. Photo reduction of CO2 is one of the promising solutions for this problem because relatively low temperature and atmospheric pressure that are required for the reaction. In this manuscript, Ali et al summarized various factors that can affect the photocatalytic reaction, namely role of organic contaminations, geometry and feed manner of the reactors, catalyst support types, intensity and frequency of incident light, temperature and also the reactant feed ratio. The fact different research groups chose different parameters makes the comparison between reported results impossible. Hence they proposed a terms of apparent quantum yield (AQY) to normalize the photocatalytic performance from different set-ups conducted in different research groups. Such efforts are meaningful in the field of gas phase photocatalytic CO2 reduction. A few typos and errors need to be fixed before the manuscript to be accepted for publication.

1), page 13, line 303-304.  Further increase in CO2 decelerated the yield, definitely not due to the extra adsorption of water, please refer to Ref. 34 for original description!

2), page 14, line 308, “absorption” should read “adsorption”.

3), Eq. 6, definition of “number of incident photons”, apparently, incident phonons should mean all the phonons that shine to the reactor but not the one expressed by equation 6. Some other terminology such as “effective number of incident photons” may be used.

4), page 12, concentration ratio (CR) need to be clearly defined for better understanding this effect.

Author Response

Response to Reviewer 1

Increasing concentration of CO2 from human activities is believed to be the reason for global warming. Photo reduction of CO2 is one of the promising solutions for this problem because relatively low temperature and atmospheric pressure that are required for the reaction. In this manuscript, Ali et al summarized various factors that can affect the photocatalytic reaction, namely role of organic contaminations, geometry and feed manner of the reactors, catalyst support types, intensity and frequency of incident light, temperature and also the reactant feed ratio. The fact different research groups chose different parameters makes the comparison between reported results impossible. Hence they proposed a terms of apparent quantum yield (AQY) to normalize the photocatalytic performance from different set-ups conducted in different research groups. Such efforts are meaningful in the field of gas phase photocatalytic CO2 reduction. A few typos and errors need to be fixed before the manuscript to be accepted for publication.

1), page 13, line 303-304.  Further increase in CO2 decelerated the yield, definitely not due to the extra adsorption of water, please refer to Ref. 34 for original description!

Response: Thank you for important comment. We have modified the explanation according to suggestion in accordance with the original description of the referred research article.

2), page 14, line 308, “absorption” should read “adsorption”.

Response: Thank you for pointing out. The correction has been done in the revised manuscript.

3), Eq. 6, definition of “number of incident photons”, apparently, incident phonons should mean all the phonons that shine to the reactor but not the one expressed by equation 6. Some other terminology such as “effective number of incident photons” may be used.

Response: Thank you for the suggestion. The word has been modified to the recommended one in the revised manuscript.

4), page 12, concentration ratio (CR) need to be clearly defined for better understanding this effect.

Response: Thank you for nice comment. We have more clearly addressed the CR definition in the revised manuscript.

Reviewer 2 Report

The review entitled “Gas phase photocatalytic CO2 reduction, “A brief overview for benchmarking”””by Ali et al., written by a well established group on the gas phase photocatalytic CO2 reduction research studies adds an important comparative analysis of the parameters, methods and techniques used in this field. The review is a good summary of the studies performed in this field and deserve publication after some modification which actually may improve the review. However, as stated in the title this is a brief overview for benchmarking thus depend on the authors what is appropriate and how long they could consider it. Some proposed comments: Please expand the conclusion section by considering briefly the advantages and disadvantages of the techniques, parameters, methods, etc. involved in gas phase photocatalytic reduction of CO2. Please add more input on the limitations of one method over the other. Please discuss on the effect of different temperatures for CH4 yield and AQY as you did for variable light intensities in figure 12. Please add in the conclusion section proposed methodology and technique to be used for best results on the co2 reduction in terms of AQY. Please add “future directions” in the conclusion section as a follow up way to use for the research teams working in this field.

Author Response

Response to Reviewer 2

The review entitled “Gas phase photocatalytic CO2 reduction, “A brief overview for benchmarking”””by Ali et al., written by a well-established group on the gas phase photocatalytic CO2 reduction research studies adds an important comparative analysis of the parameters, methods and techniques used in this field. The review is a good summary of the studies performed in this field and deserve publication after some modification which actually may improve the review. However, as stated in the title this is a brief overview for benchmarking thus depend on the authors what is appropriate and how long they could consider it.

Some proposed comments:

1). Please expand the conclusion section by considering briefly the advantages and disadvantages of the techniques, parameters, methods, etc. involved in gas phase photocatalytic reduction of CO2.

Response: Thank you for the meticulous suggestion. In the revised manuscript, we have modified the conclusions section with the changes made highlighted in yellow.

2). Please add more input on the limitations of one method over the other.

Response: Thank you for the suggestion. The information regarding the limitations of the batch reactor method as compared to continuous method has been explained more comprehensively in the revised manuscript (Please see the yellow highlighted region).

3). Please discuss on the effect of different temperatures for CH4 yield and AQY as you did for variable light intensities in figure 12.

Response: Thank you for nice comment. The discussion regarding the effect of temperature for the CH4 yield has been addressed more elaborately in accordance with the reported literature (section 6). However, the cited reference articles did not mention about the effect of temperature on AQY, but only about variable light intensities. However, in general with increase in temperature the thermal energy minimizes the kinetic barrier and thus enhances the AQY. Please see the yellow highlights mentioning the brief discussion.

4). Please add in the conclusion section proposed methodology and technique to be used for best results on the CO2 reduction in terms of AQY.

Response: Thank you for nice comment. We have added the proposed methodology in the revised manuscript.

5). Please add “future directions” in the conclusion section as a follow up way to use for the research teams working in this field.

Response: Thank you for nice comment. The future directions as a follow up way, has been added in the modified manuscript.

Round 2

Reviewer 2 Report

The authors of the review entitled "Gas phase photocatalytic CO2 reduction, “A brief overview for benchmarking”"" by Ali et al., addressed all the comments and questions after first revision. They expanded the conclusion section and included the suggested parts into the review. With the new additions the review deserves publication a it is.